# Moulding hydrodynamic 2D-crystals upon parametric Faraday waves in shear-functionalized water surfaces

Mikheil Kharbedia [1], Niccolò Caselli[1,2], Diego Herráez-Aguilar[3], Horacio López-Menéndez[1,2], Eduardo Enciso [1], José A. Santiago[1,4] & Francisco Monroy [1,2✉]

Faraday waves, or surface waves oscillating at half of the natural frequency when a liquid is vertically vibrated, are archetypes of ordering transitions on liquid surfaces. Although unbounded Faraday waves patterns sustained upon bulk frictional stresses have been reported in highly viscous fluids, the role of surface rigidity has not been investigated so far. Here, we demonstrate that dynamically frozen Faraday waves—that we call 2D-hydrodynamic crystals—do appear as ordered patterns of nonlinear gravity-capillary modes in water surfaces functionalized with soluble (bio)surfactants endowing in-plane shear stiffness. The phase coherence in conjunction with the increased surface rigidity bears the Faraday waves ordering transition, upon which the hydrodynamic crystals were reversibly molded under parametric control of their degree of order, unit cell size and symmetry. The hydrodynamic crystals here discovered could be exploited in touchless strategies of soft matter and biological scaffolding ameliorated under external control of Faraday waves coherence.

[1] Departamento de Química Física, Universidad Complutense de Madrid, Ciudad Universitaria s/n, Madrid, Spain. [2] Translational Biophysics, Instituto de Investigación Sanitaria Hospital Doce de Octubre, Madrid, Spain. [3] Instituto de Investigaciones Biosanitarias, Universidad Francisco de Vitoria, Ctra. Pozuelo-Majadahonda, Pozuelo de Alarcón, Madrid, Spain. [4] Matemáticas Aplicadas y Sistemas, Universidad Autónoma Metropolitana Cuajimalpa, Vasco de Quiroga 4871, Ciudad de México, México. ✉email: monroy@ucm.es

Nonlinear surface waves embrace a richness of hydro-dynamic behaviors dated back to Faraday, who discovered the existence of bounded oscillations at the free surface of vibrated liquids[1]. The Faraday waves (FWs) are resonant oscillations of the unbalanced liquid weight restored by the surface response, being characterized by a subharmonic response appeared as a parametrically resonant mode with frequency one half of the forcing excitation[2,3]. A critical driving force defines the onset for the Faraday instability[4], which has been prescribed as a hydrodynamic regime for ordering surface waves in the macroscopic realm[5,6]. The existence of unbounded patterns of nonlinear surface waves (stripes, square or triangular unit cell, and even quasi-crystals) at large aspect-ratio have been reported in highly viscous fluids close to the Faraday instability[7–10], where wave ordering is scaffolded upon bulk frictional stresses[10,11]. That bulky class of FW-patterns have been exploited, for instance, to organize granular layers[12], achieve colloidal lattices on top of structured suspensions[13], or in templating cell culture patterns[14,15]. Moreover, container-bounded Faraday waves have been employed to create standing patterns in inviscid liquids[16,17], in analogy with the parametric excitations of quantum condensates[18,19] and the optical lattices able to confine ultra-cold atom gases[20]. As an innovative synthesis concept in materials science, externally guided FW-patterning can be envisioned as a powerful liquid-based templating approach for material micromolding.

Prominent organizational rules are known to exist in nonlinear flows in two dimensions[21–25], in which they serve as hydrodynamic interactions for surface waves organization[26]. The identification of these subtle dynamical structures presents an ongoing challenge for adequately designing and understanding surface wave patterns. Harmonic nonlinear resonance is the key catalyst for surface wavefield ordering as it causes significant energy transfer and hydrodynamic organization among nonlinear surface waves[27]. In inviscid liquids, disordered fields of nonlinear surface waves are often identified as turbulent cascades made of harmonic modes internally created by resonant couplings at a random distribution of wave phases[21]. These disorganized nonlinear surface waves (without subharmonic response), are well-known in the regime of weak turbulence in which they display a characteristic Kolmogorov-Zakharov (KZ) spectrum[21]. Above FW-threshold, nonlinear surface waves can appear as parametrically resonant Faraday waves occurred upon oscillatory change of the natural surface response at extrinsic control by the driving force[3,5]. Being subharmonic in nature, the cascades of Faraday waves lead the bifurcation from the harmonic response[4,28]. Even stronger FW-turbulences can be found at coexistence with the ordinary nonlinear surface waves[29–31], which have been studied in the context of surface wave structuring upon dissipative vortices[32]. Arguably, intrinsic surface stiffening could harness the turbulent randomness into coherence domains able to sustain wave ordering. Despite the enormous avenues of applications that can be envisioned upon parametric control of wave freezing in liquid surfaces, the generation of stiffness-induced patterns of nonlinear surface waves have been not explored so far.

By addressing in detail the free-standing regimes of discrete resonances driven on nonlinearly interacting surface waves, our work focusses on the coherent parametric states that can entail wave ordering upon in-plane shear rigidity endorsed by an adsorption film. By taking advantage of the archetypal Faraday experiment, we systematically analyze the parametrically regulated FW-regime by using an optical probe that relied on laser Doppler vibrometry (LDV). We demonstrate the emergence of steady, free-standing patterns of Faraday waves in the shear rigidity-functionalized surfaces under no influence of wave bounding conditions even in the absence of a bulk friction. Since these novel surface patterns resemble a two-dimensional crystalline order, we call them as hydrodynamic 2D-crystals, provided a steady pattern of waves appears stable like frozen at the fluid surface.

In pursuit of discovering the role of surface stiffness in forming hydrodynamic crystals, we engineered the free surface of water by adsorbing rigid films of soluble surfactants at the air/water (A/W) interface. We employed β-aescin, which is a saponin biosurfactant with anti-inflammatory and vasoconstrictor effects[33], capable of forming rigid monolayers due to hydrogen bonding[34–37]. When Faraday waves are tuned upon external driving, hydrodynamic 2D-crystals can be molded in terms of the excitation characteristics. Since bulk friction remains as low as corresponds to the inviscid water subphase, we demonstrate reversible surface rigidization as the key-player promoting the mode freezing necessary to form the novel class of hydrodynamic 2D-crystals.

## Results

**Nonlinear surface waves excited on water surfaces.** In our search for steady, free-standing patterns of parametric nonlinear surface waves (NLSWs), we forced frequency-discretized surface wavefields under resonant liquid oscillations (see Supplementary Note 1 for a physical portrayal). To implement a NLSW-platform for hydrodynamic templating, we exploited gravity-capillary waves excited at large aspect ratio on an extensive water surface under monochromatic driving (see "Methods"). Figure 1a depicts the experimental setup used to drive and probe nonlinear surface waves upon parametric control of the liquid resonances at discretized states externally regulated by a driving force. The forcing device was designed to vertically vibrate a liquid container of large lateral dimensions (cylindrical diameter $D = 20$ cm) at a variable amplitude ($A$) and fixed driving frequency ($\omega_0$), which determine the driving acceleration as $a = A\omega_0^2$ (see "Methods"). As hydrodynamic paradigm, we considered an incompressible Newtonian fluid (water) with a density ($\rho$) and surface tension ($\sigma$) adequate to support nonlinear gravity-capillary waves[38]. We performed experiments spanning across the critical Faraday acceleration[39] ($a_F$), which tags the onset of the parametric resonance exhibiting subharmonic response at $\omega_0/2 \equiv \omega_{1/2}$, as promoted by liquid inertia (Supplementary Note 1). By focusing on the capillary domain ($\omega_0 \geq \omega_c$), the FW-threshold can be varied in terms of the kinematic viscosity ($\mu$) as[39]:

$$a_F = 8\mu(\rho/\sigma)^{1/3}(2\pi\omega_0)^{5/3} \qquad (1)$$

for given values of the density to surface tension ratio ($\rho/\sigma$). An extension to the complete domain can also be made by using the gravity-dependent effective tension $\sigma_{eff} = \sigma(1 + \rho g \lambda^2/4\pi^2\sigma)$ defined in terms of the mode wavelength $\lambda$ (see "Methods").

Equation (1) was validated for pure water and the other surfaces considered in this work (see Supplementary Tables 1 and 2 and Supplementary Fig. 1). As a reference frequency, we choose $\omega_0 = 47$ Hz, for which Eq. (1) predicts $a_F \approx 2$ m s$^{-2}$ in water at room temperature ($\sigma = 0.072$ Nm$^{-1}$, $\rho = 10^3$ kg m$^{-3}$ and $\mu \approx 2$ m$^2$ s$^{-1}$). This arbitrary choice lies in the capillary wave (CW) domain ($\omega_0 \gg \omega_c \approx 14$ Hz), where free-standing FWs propagate in the deep-water regime as NLSW-ensembles of large aspect-ratio with wavelengths shorter than the vessel dimensions ($\lambda_0 \approx 6$ mm $\ll D$) (Fig. 1b; see "Methods"). Similar results were obtained for pure gravity waves (GWs) at frequencies below (but close) to $\omega_c$ (see Fig. 1b top panel and Supplementary Table 1). Using this hydrodynamic platform, a wide NLSW-scenario was explored in terms of a reduced driving acceleration $\Gamma \equiv a/g$; hereinafter, the control parameter $\Gamma$ will be referred to the dimensionless Faraday threshold for pure water ($\Gamma_F \equiv a_F/g \approx 0.2$).

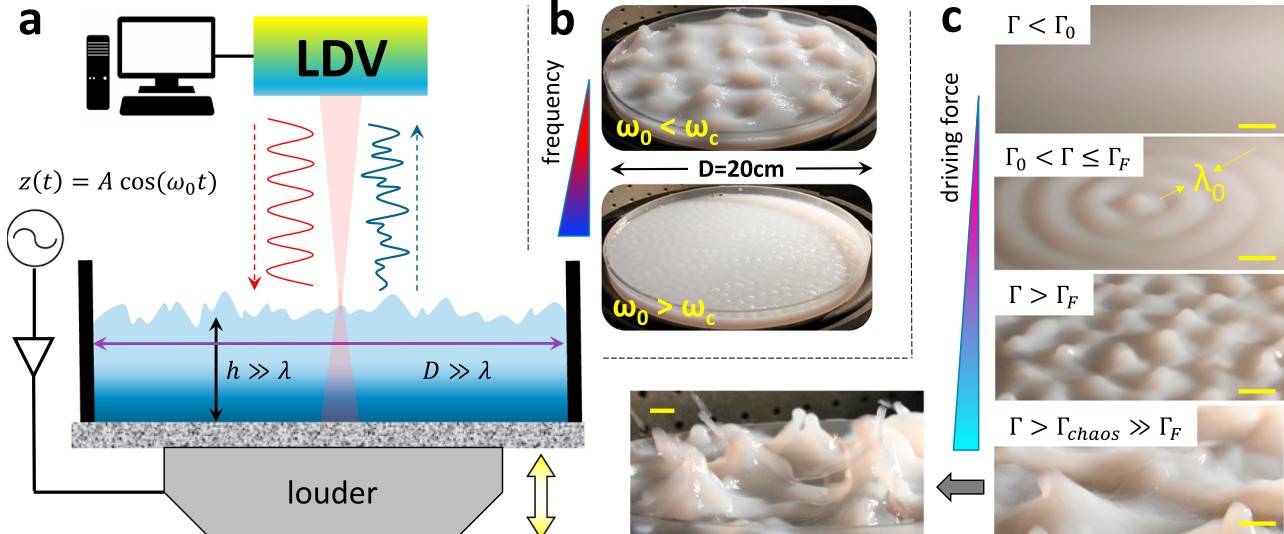

**Fig. 1 Inducing and detecting surface waves. a** Schematics of the experimental setup. Surface waves were generated at the A/W-interface under vertical agitation in a cylindrical container with large aspect-ratio (diameter $D = 20$ cm and height $h = 2$ cm, both larger than the typical induced oscillation wavelength $\lambda$). The container was attached to an electromechanical driver (louder) that induced a periodic vertical displacement $z(t)$ at frequency $\omega_0$ and amplitude $A$. The surface wavefield displacement was recorded as a function of time by means of a Laser Doppler Vibrometer (LDV) (see "Methods"). **b** Images of the investigated surface for water mixed with a low concentration of colloidal particles (PS-MMA) acting as white colorant (see "Methods"), obtained when the vessel was vertically vibrated at increasing frequency (from red to blue) in the gravity ($\omega_0 < \omega_C$) and capillary regime ($\omega_0 > \omega_C$), respectively. **c** Images of characteristic surface response for increasing driving force (from red to blue), reported in terms of the reduced acceleration $\Gamma \equiv A\omega_0^2/g$, in the capillary wave regime ($\omega_0 = 47$ Hz, hence $\lambda_0 = 5.9$ mm; see "Methods"). From top to bottom panel: low amplitude ($\Gamma < \Gamma_0$) exhibiting a linear response; intermediate amplitude ($\Gamma_0 < \Gamma < \Gamma_F$) exhibiting concentric waves; amplitude above the Faraday threshold ($\Gamma > \Gamma_F$) displaying surface scrambled ripples; extremely high amplitude ($\Gamma > \Gamma_{chaos} \gg \Gamma_F$) where chaotic waves give rise to droplet ejections (see the bottom left panel with enlarged view). The scale bar is 5 mm.

Figure 1c displays the waving textures captured in the relevant amplitude domains for increasing driving force. Below a Hookean limit ($\Gamma < \Gamma_0$), because the low amplitude CWs are invisible to the naked eye (see Fig. 1c top panel), LDV detected the linear surface response as a monochromatic transversal wave appeared at $\omega_0$ (see Supplementary Fig. 2). At moderate driving force ($\Gamma_0 < \Gamma < \Gamma_F$), we observed high-amplitude surface undulations as circular waves mastered by the fundamental wavelength $\lambda_0$ (see Fig. 1c second panel). Further excitation above the Faraday onset ($\Gamma > \Gamma_F$), caused progressive surface roughening as scrambled ripples of variable amplitude, which were associated to the FW-parametric resonance (see Fig. 1c third panel). Finally, at very high driving forces ($\Gamma > \Gamma_{chaos} \gg \Gamma_F$), increasing waving disorder gave rise to a dynamically chaotic surface (see Fig. 1c bottom panel). At $\Gamma \gg \Gamma_{chaos}$, the surface became unstable even observing ejection of droplets (see outset in Fig. 1c; bottom). These amplitude domains were achievable not only with CWs of variable frequency but also with GWs below $\omega_c$ (see Supplementary Fig. 3). All these NLSW-states were robustly reproduced in water and in other liquids, however, we found that the transitions between them are strongly dependent on the liquid viscosity (see Supplementary Table 1).

Because the low viscosity of water warrants nonlinear surface waves propagation at moderate wave damping[38,39], but is not sufficiently high for sustaining free-standing surface wave patterns built upon bulk friction[15], the results displayed in Fig. 1 validate our dynamic platform for unbounded nonlinear surface waves both in the gravity and capillary domain. Furthermore, in the Faraday regime, at $\Gamma > \Gamma_F$, we assured the fulfillment of the large aspect-ratio condition[40], and prevented undesired stationary waves upon edge bounding.

**Coherent and incoherent cascades of nonlinear surface waves.** Figure 2a shows the spectral features determined by laser Doppler

vibrometry in the NLSW-states spanned by varying $\Gamma$ with respect to $\Gamma_F$ (see "Methods"). At low driving force ($\Gamma = 0.1 < \Gamma_F$; Fig. 2a, first row), the nonlinear surface response emerged as a cascade of discrete harmonics exhibiting discrete peaks at multiple integers of the fundamental frequency (at $\omega_n = n\omega_0$, with amplitudes $\psi_n(\omega_n)$; see "Methods"). The recorded cascade follows the KZ-decay, i.e., with power spectral density (PSD $\equiv f t_n^2 \psi$) $\sim \omega^{-17/6}$, characteristic for weak CW-turbulence[21,41]. A homogenous distribution of the mode phases was observed (see Fig. 2b, first panel), which confirmed the natural phase randomness of the nonlinear surface waves created in the KZ-regime at $\Gamma < \Gamma_F$. At the onset of the Faraday instability ($\Gamma = 0.2 \approx \Gamma_F$; Fig. 2a, second row), LDV revealed the emergence of the subharmonic peak ($\omega_{1/2} \equiv \omega_0/2$), in addition to the fundamental mode ($\omega_0$). Moreover, two super-posed cascades decay alternating within the lower harmonics up to a Faraday cut-off ($n \lesssim n_F \approx 20$). They correspond, respectively, to the ordinary KZ-cascade of the fundamental mode ($o \equiv$ KZ at $n\omega_0$), and to the extraordinary cascade of the Faraday sub-harmonic ($e \equiv$ F at $n\omega_{1/2}$). Both cascades show similar decay rates as PSD $\sim \omega^{-5}$, consistent with cooperative scaling characteristic of strong, highly correlated, wave turbulence[32]. Interestingly, those composite FW-cascades cease at $n_F^{(max)}$, which correspond to the highest mode before incoherent KZ-turbulence became dominant. Indeed, for $n \geq n_F^{(max)}$, the capillary-like $\omega^{-17/6}$-decay remerged (see also Supplementary Fig. 4). This KZ-Faraday (KZF) hybridization was only detected ca. $\Gamma \gtrsim \Gamma_F$, where phase-locking was still weak (see Fig. 2b, second panel). Above the Faraday onset ($\Gamma \geq \Gamma_F$; Fig. 2a, b, third row), the composite ($o \cup e$) cascade appeared as a unified FW-cascade decaying as PSD $\sim \omega^{-5}$. In this case, two mode-coupling features were detected; these are peak broadening indicating inter-cascade energy exchange ($o \leftrightarrow e$), and phase-locking leading to intermodal coherence. Hereinafter, we will refer to this class of $\omega^{-5}$-decaying cascade as pure-FWs (see also

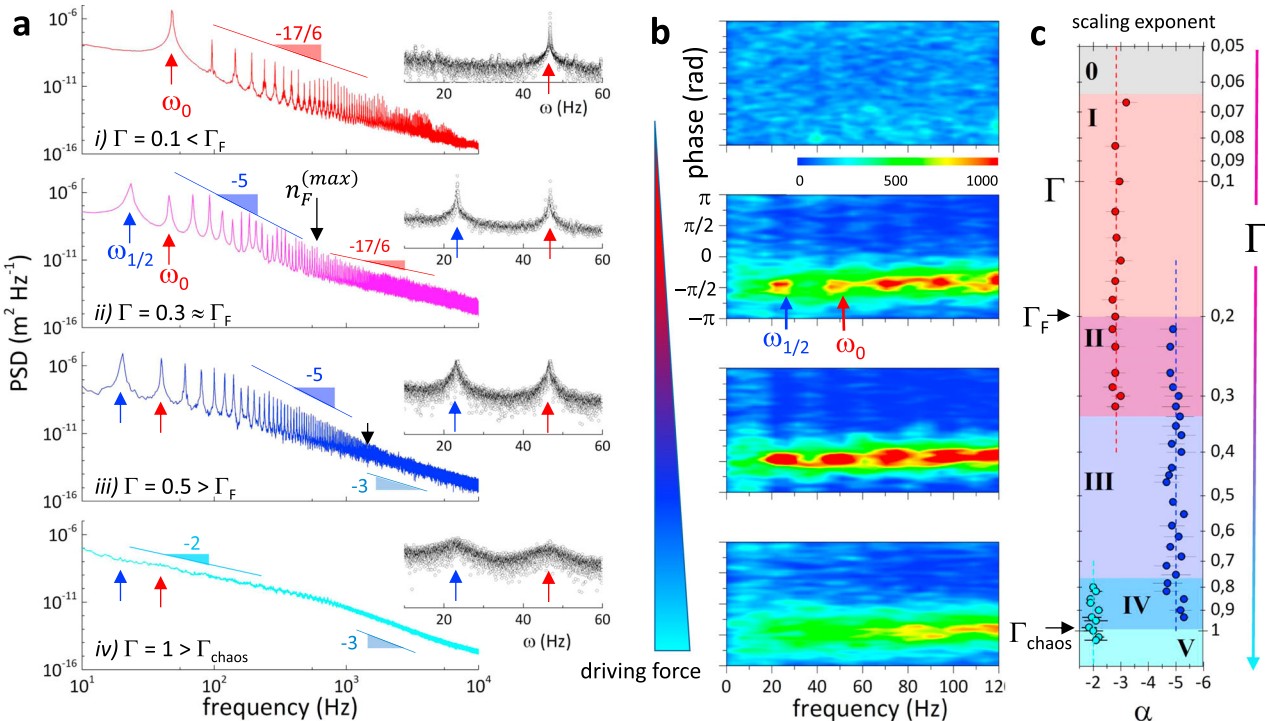

**Fig. 2 Spectral power cascade and coherence of nonlinear surface waves. a** Power spectral density (PSD) of the NLSWs excited at the air/water interface by vertically oscillating the vessel at frequency $\omega_0 = 47$ Hz for increasing acceleration $\Gamma$; from top to bottom: (i) Kolmogorov-Zakharov (KZ) spectrum of weak wave turbulence (at $\Gamma < \Gamma_F$) with $\omega^{-17/6}$-intensity decay typical of cascades of nonlinear capillary waves; (ii) hybrid spectrum of turbulence with superimposed subharmonic $\omega^{1/2}$ (blue arrow), and fundamental $\omega_0$ (red arrow) cascades (at $\Gamma \geq \Gamma_F$). The Faraday $\omega^{-5}$-cascade predominates in the lower frequency range, while the $\omega^{-17/6}$-decay prevails at higher frequencies. (iii) Pure Faraday spectrum of inertia-controlled resonance with $\omega^{-5}$-envelope that ends into a terminal $\omega^{-3}$-tail (at $\Gamma > \Gamma_F$); (iv) chaotic continuous spectrum at crossover between Landau's turbulence $\sim\omega^{-2}$, and terminal frictional death $\sim\omega^{-3}$ (at $\Gamma \gg \Gamma_F$). The insets highlight the spectral range of $\omega_0$ and $\omega_{1/2}$ in linear scale. Numerical simulations and theoretical interpretations of these spectra can be found in Supplementary Note 1. **b** Phase distribution of the spectra reported in (**a**) for increasing driving force (from red to blue). Phase locking between the cascade resonances is observed when $\omega_{1/2}$ emerges. **c** Scaling exponent ($\alpha$) that fitted the peak cascades as a function of $\Gamma$, highlighting the transitions between the regimes described in (**a**); they are qualitatively delimited by different color areas. The gray (0) area represents the linear regime; pink (I) area the $\omega^{-17/6}$ KZ-decay; the purple (II) area the hybrid cascade (KZF); the dark blue (III) area the pure Faraday cascade ($\omega^{-5}$); the blue (IV) area the coexistence of Faraday and $\omega^{-2}$-decay cascades of Landau's unsteadiness; finally, the light blue (V) area delimits the completely chaotic cascades. The error bars correspond to the fit uncertainty.

Supplementary Fig. 5). Previous experimental works identified the $\omega^{-5}$-spectral envelope as a phenomenological FW-distinctive compatible with the parametric resonance of the surface response[29], which can be interpreted as a dominance of bulk inertia in promoting the wavefield subharmonic resonance at balance with the driving source (see Supplementary Note 1). The FW-cascades made of harmonics and subharmonics were observed to dead in a terminal domain (see also Supplementary Fig. 5), which is compatible with a $\omega^{-3}$-frictional background (Supplementary Note 1).

Beyond the Faraday threshold ($\Gamma \gg \Gamma_F$; Fig. 2, fourth row), we observed a transition towards a chaotic regime at $\Gamma > \Gamma_{chaos} \approx 0.9$. In this domain, the observed cascades were characterized by a Lorentzian decay as PSD $\sim \omega^{-2}$, peak broadening, and phase decoherence followed by frictional death at higher frequencies ($\sim\omega^{-3}$), which evidenced the chaotic nature of the surface wave unsteadiness appeared at developing continuous turbulence[42]. Since the spectra did not show discrete resonances but rather became a frequency superposition (see Supplementary Fig. 6), we refer to those NLSWs as continuous chaotic spectra by reference to the Landau's conjecture for turbulent flows[42]. A global theoretical characterization of these NLSW regimes, including cascade simulations and wavefield analytics for the spectral exponents, was performed in terms of equivalent nonlinear oscillator (NLOs) (see Supplementary Note 1).

To highpoint the transitions between the aforesaid NLSW-states, Fig. 2c shows the fitted values of the scaling exponents PSD $\sim \omega^{-\alpha}$ as spanned in terms of six regimes (labeled from 0 to V; see caption). Pure Faraday waves underlying strong resonant coupling and phase coherence were found in a relatively broad domain of driving forces (state III: $1.5\Gamma_F \lesssim \Gamma \lesssim 4\Gamma_F$). This genuine FWs-domain is flanked by hybrid regimes contaminated by decoherent interactions. Similar sequences of waving states were observed at different driving frequencies either in water or with other liquids (see Supplementary Fig. 7), which suggests a universal NLSW-behavior as depicted in the state diagram of Fig. 2c. By invoking a mechanical analogy between NLSW fields and forced NLOs both under parametric control of discretized waving states upon monochromatic driving (see Supplementary Note 1), we portrayed a unified picture with the concept of wavefield self-interaction at the core of the coherent resonances observed in the Faraday domain (see Supplementary Note 2).

**Hydrodynamic crystals on shear-functionalized fluid surfaces.** After having characterized the NLSW-states, here we will deliver the main outcome of this work: the induction in the coherent Faraday waving state of large hydrodynamic 2D-crystals after inducing surface-shear rigidization. As an efficient stiffening

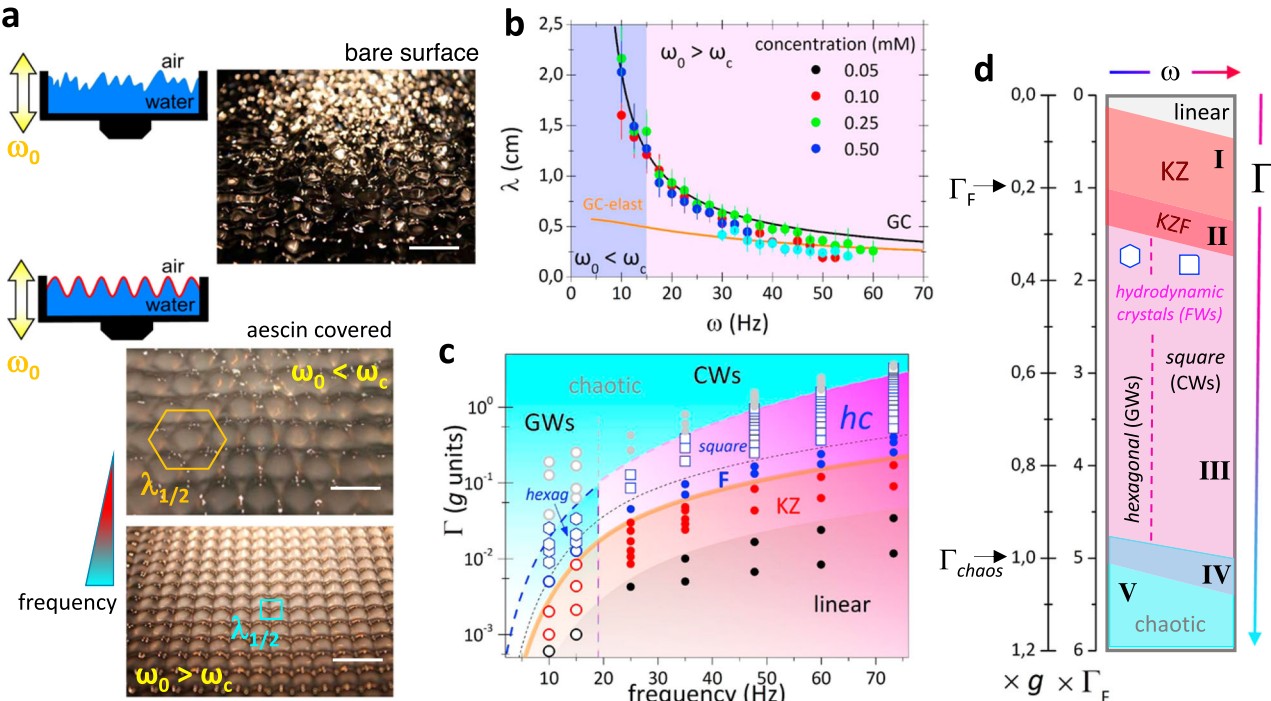

**Fig. 3 Hydrodynamic crystal formation and dispersion relation. a** Schematics and real images of the bare water surface (top) and for the water surface covered with β-aescin (bottom), under the same parametric excitation in the pure Faraday regime. Two distinct arrangements can be tailored with increasing frequency (from red to blue): hexagonal packing in the gravity wave regime ($\omega_0 < \omega_c$), and square packing in the capillary wave regime ($\omega_0 > \omega_c$). The ordering is induced by a rigid surface functionalization due to adsorption of soluble β-aescin monolayers (solid-like at cmc ≈ 0.4 mM). **b** Hydrodynamic crystal dispersion relation measured for different aescin concentration (solid dots), along with theoretical dispersion predictions for pure gravity-capillary waves ($\sigma \approx 30$ mN m$^{-1}$; black line), and elasticity-controlled gravity-capillary waves ($G/\sigma \approx 30$; orange line, see Supplementary Note 5). The wave characteristics correspond to the Faraday subharmonic, i.e., $\omega \equiv \omega_{1/2} = \omega_0/2$ and $\lambda \equiv \lambda_{1/2}$. The error bars correspond to the uncertainty in determining the lattice period $\lambda_{1/2}$ from the optical images of the exciting surface. **c** State diagram in the $\Gamma - \omega$ space for parametrically nonlinear surface waves in water functionalized with aescin. Linear (black dots), KZ (red dots), Faraday (F; blue dots) and chaotic (gray dots) spectral cascades are identified, along with the emerging states corresponding to hydrodynamic crystals (hc), and their geometry reported as white hexagons and squares, respectively. The distinct regimes are qualitatively delimited by lines and color areas. Note that hydrodynamic crystals occur in the pure Faraday regime slightly above $\Gamma_F$. **d** Simplified schematics of the state diagram evaluated in (**c**). A comprehensive interpretation in terms of wave interactions and symmetries is given in Supplementary Note 3.

agent, we employed β-aescin reversibly adsorbed as a rigid monolayer at the A/W-interface. The surfactant was dissolved at a concentration close to its critical micellar concentration (cmc ≈ 0.4 mM). Then, we excited nonlinear surface waves by forcing the hydrodynamic states to lie in the pure Faraday domain. Figure 3a compares the textures of Faraday waves observed under parametric excitation either in a bare water surface (top panel) or in a covered surface functionalized with β-aescin (bottom panel). The occurrence of a coherent surface wave freezing induced by the presence of the surfactant is remarkable since the equivalent bare surfaces appeared completely disordered under identical waving excitation. As a matter of fact, only the pure Faraday waves could arrange into stationary patterns with a long-range order forming a macroscopic 2D-crystal (see Supplementary Fig. 8). Theoretical analyses with NLOs under stiffening revealed the key role of rigidity-guided self-focusing as a main cause for wave freezing (see Supplementary Note 1).

Notably, crystal symmetry can be harnessed depending on the dispersion regime (see Fig. 3a, bottom panel). By tuning the driving frequency $\omega_0$ with respect to the capillary frequency $\omega_c$, we obtained either hexagonal crystals with the triangular symmetry in the gravity regime ($\omega_0 < \omega_c$), or square crystals in the capillary regime ($\omega_0 > \omega_c$). The size of the unit cell was fixed by the FW-subharmonic wavelength $\lambda_{1/2} \equiv 2\lambda_0$, which varies with $\omega_{1/2}$ ($\equiv \omega_0/2$) as determined by the gravity-capillary dispersion

equation (for given $\sigma$ and $\rho$ under nonlinear corrections due to surface elasticity $G$; see Fig. 3b and Supplementary Note 5).

These quantitative results, complemented with the graphical evidence in Supplementary Fig. 8, prove that the crystalline structure can be moulded under parametric control (by varying $\Gamma$ at fixed $\omega_0$). Figure 3c reports the state diagram found in water surfaces covered by β-aescin, as determined by the systematic cartography of the $\Gamma - \omega$ space. Figure 3d shows that although Faraday waves exist in a broad interval of driving acceleration ($\Gamma_F < \Gamma < \Gamma_{chaos}$), hydrodynamic crystals with a long-range order were only achieved well inside the pure Faraday domain of the covered surface (state III; Fig. 3d). The upper and bottom limits of this domain of frozen patterns remain almost unaltered with respect to its homologous state in the bare surface (see Fig. 2c). Disordering transitions can be understood as wavefield defocusing effects tending to disentangle the rigidity skeleton that supports the FW-patterns in the coherence domain ($\Gamma_F < \Gamma < \Gamma_{chaos}$); either by losing resonant inertia (in the KZ-state at $\Gamma < \Gamma_F$), or by increasing frictional unsteadiness (Landau chaos at $\Gamma > \Gamma_{chaos}$) (see Supplementary Note 2). In view of theoretical NLO equivalents with symmetries broken by surface softening and/or increasing friction, the NLSW state diagram has been argued to share a common core of self-focusing interactions grounded on bulk inertia and surface elasticity (see Supplementary Note 3).

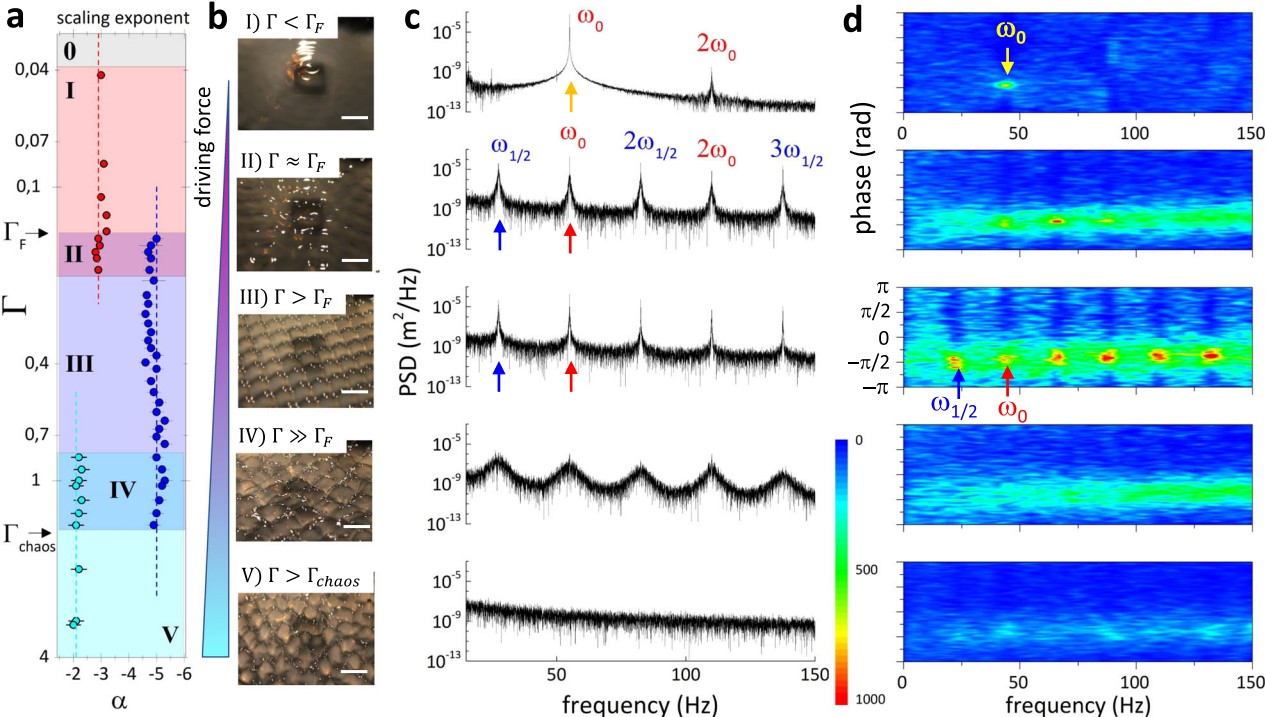

**Fig. 4 Hydrodynamic crystals: structured cascades and wave coherence. a** Scaling exponent ($\alpha$) obtained from the complete spectral cascades (see Supplementary Fig. 9) for increasing driving force (from red to blue), highlighting the transitions between the waving states qualitatively delimitated by different color areas (as in Fig. 2c). The error bars correspond to the fit uncertainty. **b** Imaging of the nonlinear surface waves at the center of the vessel for different $\Gamma$, corresponding to the subsequent spectral analyses performed, and at $\omega_0 = 47$ Hz in the capillary regime. **c** Power spectral density (PSD), and **d** phase distributions at increasing amplitude (spectral zooming at the fundamental modes and first harmonics; from top to bottom): (I) Weak KZ-turbulence ($\Gamma < \Gamma_F$) exhibiting only the harmonic cascade, small coherence between the peaks and a disordered surface. (II) Hybrid KZF-cascade at the onset of the Faraday regime ($\Gamma \approx \Gamma_F$), where the subharmonic cascade at $\omega_{1/2}$ emerged, the peaks acquired a higher degree of coherence and the surface displayed a waving pattern but still not ordered. (III) Pure Faraday waves in the regime $1.5\Gamma_F \lesssim \Gamma \lesssim 4\Gamma_F$, where the peaks narrowed, exhibiting a strong coherence, and the Faraday waves froze as a stable square pattern. (IV) For increasing acceleration ($\Gamma_F \ll \Gamma < \Gamma_{chaos}$) the peaks broadened and decoherence broke up the crystal. (V) In the complete chaotic regime ($\Gamma > \Gamma_{chaos}$), nonlinear surface wave-spectra did not show any peak, the phase distribution was random, and the waving surface became strongly disordered.

**Coherent Faraday wave freezing occurs upon phase locking**. To scrutinize the hydrodynamic skeleton underlying the 2D-ordering in FW-patterns, we identified coherent couplings in the templating NLSW-cascades. The relevant conclusions are emphasized in Fig. 4 focusing on capillary waves, although similar qualitative conclusions were drawn for gravity waves. As shown in Supplementary Figs. 8 and 9, the surfaces covered with β-aescin followed the same sequence of NLSW-states observed in the bare water surface (see Fig. 2c), exhibiting similar spectral decay scaling for all the five regimes (see Fig. 4a). The presence of ordered crystals was uniquely detected in the regime of genuinely coherent FWs (Fig. 4b and Supplementary Fig. 8). In this case the pure-FWs resonant harmonics were characterized by extreme spectral narrowing (Fig. 4c), and highly localized phase-locking (Fig. 4d). However, phase decoherence elicited resonance narrowing and progressive surface disordering. Even though, some degree of deterministic organization remained in the chaotic V-state at $\Gamma > \Gamma_{chaos}$ (see Fig. 4; bottom panels), where phase coherence persisted higher than in the genuinely disordered I-state at $\Gamma < \Gamma_F$ corresponding to incoherent KZ-turbulence (Fig. 4; top panels). These findings agree with the theoretical description of a rigidity-driven self-focused wavefield introduced to interpret the phase-locking observed in the FW-patterns (see Supplementary Note 1 and 2). A preliminary formulation of a field theory for the hydrodynamic crystalline state was inspired on these materials, from which a structural picture arose as frozen Faraday waves with a rigidity skeleton scaffolded on the discretized pieces of space and time

engendered by coherent inertial resonance with the monochromatic driving source (see Supplementary Note 3).

**Material relationship**. By exploiting experimental interfacial rheology (see Methods and Supplementary Note 4), we performed a quantitative analysis of the degree of waving ordering by reference to the in-plane shear modulus ($G$). Figure 5a shows the results for different surfactant additives. For β-aescin monolayers, we found $G$ increasing with concentration (up to $G \approx 1\,\mathrm{Nm^{-1}}$ as highest at $c \geq 2$ cmc). This significant surface stiffening elicited a progressive intensification of the degree and spatial range of the crystalline order on the hydrodynamic crystal (see Fig. 5b). This paradigmatic stiffener was compared with other surface functionalizations that did not induce enough rigidity to support NLSW-patterning, such as soluble CTAB forming fluid monolayers ($G \approx 0$)[43], and insoluble DPPC at the gel phase of monolayer packing ($G \approx 10^{-4}\,\mathrm{Nm^{-1}}$)[44]. Among the tested surfactants, only β-aescin was able to harness Faraday waves as 2D-hydrodynamic crystals, whereas the other surface monolayers did not support ordered patterns (see Fig. 5c; left panels). The efficiency of β-aescin as a 2D-rigid fabric to freeze Faraday waves is remarkable since, under the same conditions, the bulky skeletons scaffolded on the 3D-frictional stresses of colloidal solutions were unable to support hydrodynamic crystals despite their higher dimensionality (see Fig. 5c; right panels).

Furthermore, LDV-spectral analysis was exploited to quantify the intrinsic degree of hydrodynamic ordering achieved by the

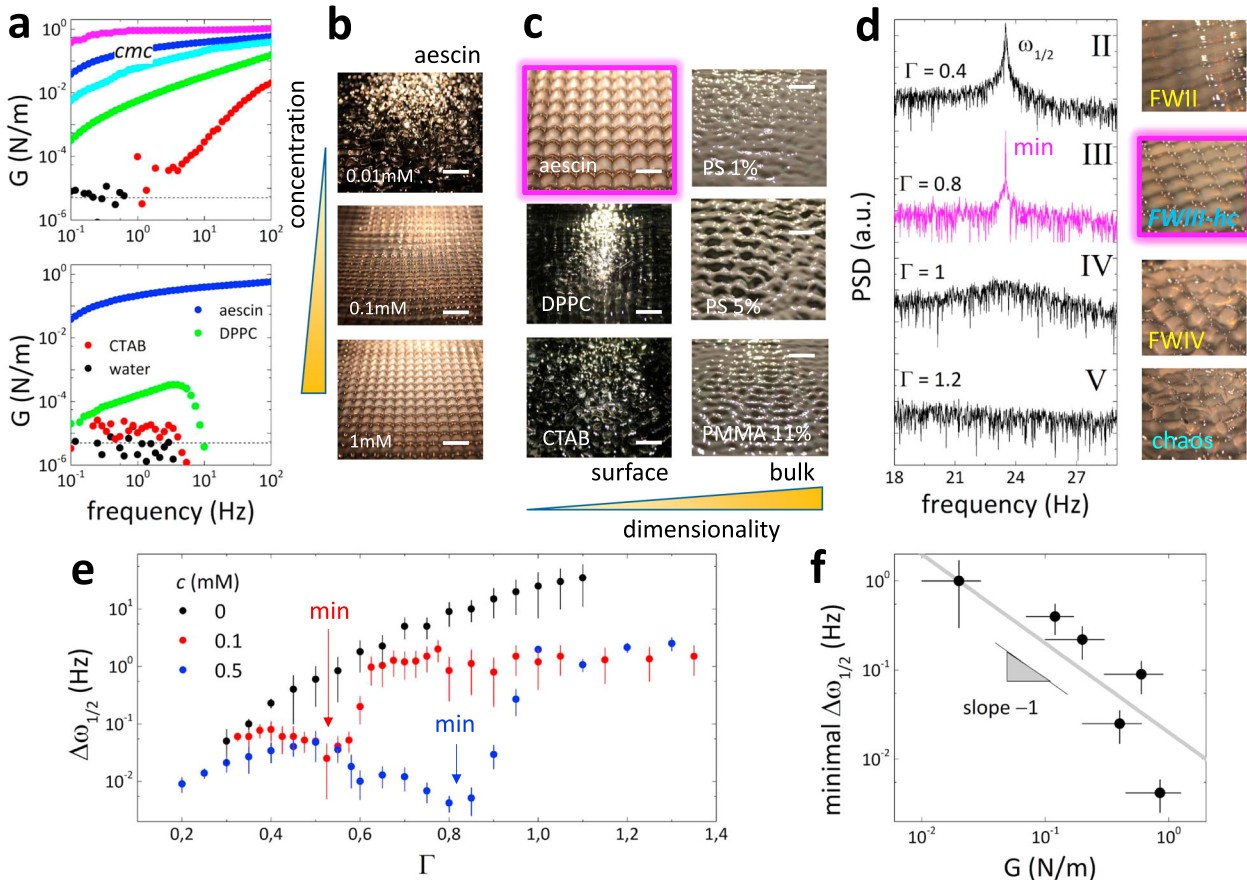

**Fig. 5 Hydrodynamic crystals: surface stiffening and spectral narrowing. a** Linear interfacial rheology (see Supplementary Note 3) of aqueous solutions for different aescin concentration (top panel), and different surfactants as CTAB and DPPC (lipid) at equivalent concentration as the surface tension of the aescin solution at cmc = 0.4 mM ($\sigma \approx 30$ mN m$^{-1}$; lower panel). The surface stiffness ($G$) varies by several orders of magnitude from the highest saturation value $G \approx 1$ Nm$^{-1}$ at 1 mM concentration ($\approx$2.5 cmc; magenta dots), 0.4 mM ($\approx$; blue dots), 0.1 mM (cyan dots), 0.05 mM (green dots) down to 0.01 mM (red dots). The black dots correspond to the bare water surface, which are compatible with the instrumental sensitivity (ca. 5 μN m$^{-1}$). **b** Surface waving images at different aescin concentration (lateral yellow bar height is proportional to the concentration) under driving frequency $\omega_0 = 47$ Hz and acceleration $\Gamma > \Gamma_F$, which both allow the formation of square-lattices of hydrodynamic crystals at $G \geq 0.1$ Nm$^{-1}$, corresponding to aescin concentrations $c \geq 0.1$ mM $\approx 0.25$ cmc. **c** Surface waving images with aescin at $c \approx$ cmc leading to hydrodynamic patterns, and two surfactant additives (CTAB and DPPC) unable to harness surface ordering at equivalent surface tension ($\sigma \approx 30$ mN m$^{-1}$). The right panels compare the disordered waves obtained with concentrated colloidal suspension entailing a bulk frictional skeleton. The bottom yellow bar height is proportional to the system dimensionality, from 2D to 3D. **d** Subharmonic peak $\omega_{1/2}$ and corresponding surface image for different $\Gamma$ in β-aescin solutions at $c \approx$ cmc (the different FW-states were tested; from II to V). The hydrodynamic crystals in the pure Faraday state show the maximum structural stability characterized by the minimum peak broadening (at $\Gamma = 0.8$; magenta line). **e** Full-width at half-maximum of the subharmonic peak ($\Delta\omega_{1/2}$) as a function of $\Gamma$ for the bare water surface (black dots), and A/W interfaces functionalized with aescin at 0.1 mM (red dots) and 0.5 mM (blue dots). The minimum values determine the maximal crystal ordering. The error bars correspond to the uncertain in the peak linewidth obtained as the standard deviation over 5 measurements. **f** Minimum value of $\Delta\omega_{1/2}$ in (**e**) as a function of aescin concentration, reported in terms of $G$. The vertical and horizontal error bars are due to the estimation of the minimum value of $\Delta\omega_{1/2}$ and to the uncertain in the rheological experiment reproducibility. The solid line with the expected linear dependence between crystal order and shear rigidity represents a guide to the eye; for higher $G$ the minimum of $\Delta\omega_{1/2}$ decreases, which corresponds to a more stable and ordered hydrodynamic crystal upon stiffening the surface skeleton with increasing aescin concentration.

aescin-supported hydrodynamic crystals. We focused on the structural peak of subharmonic resonance ($\omega_{1/2}$), which determined the size of the unit cell ($\lambda_{1/2}$) (see Fig. 3a; bottom panels). The peak sharpening observed in $\Delta\omega_{1/2}$ can be interpreted as a signature for increasing wave spatial correlations and steadiness under hydrodynamic ordering imparted by surface stiffening ($\Delta\omega_{1/2} \to 0$ for ideal hydrodynamic crystals; similarly to the peak sharpening of Rayleigh-scattering in highly correlated systems[45]. Figure 5d shows $\Delta\omega_{1/2}$ observed for increasing $\Gamma$ in aescin-supported FW-patterns. At the beginning of the pure Faraday regime, where incipient crystalline ordering began to emerge (Fig. 5d; right images), a finite broadening ($\Delta\omega_{1/2} > 0$ at $\Gamma \approx 2\Gamma_F$) was detected. The sharpest peak corresponded to the maximal

degree of crystalline ordering (ca. $\Gamma \approx 4\Gamma_F$), still within the pure-FW domain. Further crystal distortion elicited progressive peak broadening ($\Delta\omega_{1/2} \gg 0$ at $\Gamma \approx 5\Gamma_F$), which was followed by hydrodynamic disordering until macroscopic melting in the chaotic regime ($\Delta\omega_{1/2} \to \infty$ at $\Gamma \approx 6\Gamma_F > \Gamma_{chaos}$). Figure 5e evidences the incremental effect of surface rigidization upon the intrinsic hydrodynamic ordering. Whereas a monotonic increase of $\Delta\omega_{1/2}$ with $\Gamma$ was observed for the bare water surface, increasing β-aescin concentration caused progressive peak narrowing within the pure Faraday wave interval. Remarkably, the most ordered crystals persisted to stronger excitations at higher aescin concentration (see caption in Fig. 5e). The material relationship is quantitatively evidenced in Fig. 5f, which points

out a phenomenological law $\Delta\omega_{1/2} \sim G^{-1}$. Analogous to the linear trade-off between increasing shear modulus with decreasing the density of defects and structural peak broadening in bulky solids[45], our finding underlies the director role of surface rigidity in promoting the order in hydrodynamic crystals.

## Discussion

We proved the existence of free-standing hydrodynamic 2D-crystals as macroscopic patterns of unbounded Faraday waves reversibly created as coherent excitations in water surfaces functionalized with a stiffening agent. From a synthetic stand-point, these hydrodynamic crystals can be molded as FW-patterns in terms of two control parameters: excitation frequency and wave amplitude. Their crystalline structure was determined by the surface characteristics, which stem, exclusively, from the NLSW dispersion regime and from the high surface shear rigidity. By contrast to the conventional FW-patterns sustained upon bulk friction[46], and the container-dependent patterns of bounded waves[14,15,17], the new class of hydrodynamic crystals appeared under bulk material conditions that, without surface functionalization, would not allow the long-range order and the large crystal size achieved in our synthesis. The degree of crystalline order was determined by the shear stiffness: as a rule of thumb the higher $G$ the lower the number of defects that distort the structure.

Once demonstrated the ordering nature of the two-dimensional elastic stresses that freeze the FW-pattern leading to unbounded standing waves, a fundamental question came to mind: What does constitute the hydrodynamic skeleton that is required to organize such an ordered FW-patterning? Our answer was immediate from the evidence raised: crucially, it is the mode coherence within the Faraday wavefield that instates wave dis-order into organization. Since temporal coherence enabled a hydrodynamic scaffold within the considered turbulent FW-cascades[29], it must persist stationary for a time longer than the fast energy exchange between the constituting modes[26–28]. Our experiments revealed crystal formation only upon coherent FW-turbulence exhibiting strong phase-locking between harmonic and subharmonic cascades materially harnessed by a skeleton of surface shear stresses enabled by a rigid adsorption monolayer. Theoretical analytics pointed out liquid inertia at resonance with the driving force as the hydrodynamic infrastructure that divides the surface wavefield into discretized pieces of time and space with the potential to become organized. These analyses have also revealed that additional wavefield self-focusing is required for freezing into hydrodynamic crystals with a permanent form; such a phase-coherence is realized on the additional surface skeleton harnessed upon enough shear stiffening. Therefore, the Faraday waves supported hydrodynamic crystals here discovered consisted of both matter and waves, the former being guided by the later like in the de Broglie's picture of matter waves[47–49]. In a first step towards a forthcoming field theory of the new hydrodynamic crystals, a preliminary depiction of this duality has been also delivered as a classical manifestation of the spatiotemporal dis-cretization imparted under monochromatic driving. Wavefield discretization upon bulky inertial resonance and wavefield self-focusing under surface stiffening might be both considered as the main potential ingredients of a Lifshitz-type action generator for the equivalent particle-wave interaction that describes the hydrodynamic crystal in analogy with the ordered phases in correlated systems[50].

From experiments and theory, surface stiffening was revealed as the crucial material ingredient to form hydrodynamic 2D-crystals from free-standing Faraday waves. A necessary condition for FW-freezing is the adsorption of monolayers of soluble β-aescin which endowed sufficient lateral rigidity for harnessing the wave coherence (field self-focusing) needed to sustain steady FW-patterns whilst excitation remains active. Aescin belongs to the group of saponins, which are biocompatible surfactants derived from glicoterpenoid compounds that endow optimal amphiphilicity due to a high packing efficiency[51,52]. Other sur-factants like β-aescin form rigid monolayers by reversible adsorption from aqueous media[28,36], could be also used to this purpose. The molecular ingredient of our hydrodynamic crystals adsorbs spontaneously covering the whole surface, thus β-aescin, or similar relatives, should be efficient to form hydrodynamic crystals in all scales from macro to nano. Since the rigid mono-layers of these (bio)surfactants self-assemble in solution[36,52], and enable interactions with biological membranes[36,37], unprece-dented routes in (bio)material synthesis can be envisioned as an exploitation of the new hydrodynamic crystal concept.

In a panoramic view, our discovery challenges the classical intuition regarding the ability of liquid waves to mimic solid-like structures without the need of any previous phase transition and practically on demand. Undoubtedly, the finding of hydro-dynamic 2D-crystals based on a completely liquid phase of matter provides a novel counterintuitive phenomenon that opens new possibilities to rethink the role of surface waves to manipulate matter.

As an outlook, the novel hydrodynamic crystals could be immediately translated into engineering and biological applica-tions requesting structural encoding via a guiding wave field. The moulding versatility under external control makes our method meaningful for rescaled implementation in substrate-guided approaches to the synthesis of macro- to nano-structured (bio) materials[53,54]. Boosted by nanotechnological requests to mimic condensed-matter systems[55,56], our straightforward FW-platform could embed functional objects into ordered lattices with an externally tunable size and symmetry. Furthermore, the study of topological defects in hydrodynamic liquid crystals will open the possibility to study the formation of currents to drive movements and directional transport[57].

In a more fundamental perspective, the crystalline patterns of Faraday waves could result into a wave–crystal association con-ceptually linked to the de Broglie's duality for extended systems in two dimensions[47,58]. Indeed, Faraday waves have been argued as the classical paradigm of de Broglie duality[47], in which the physical nature of the guiding FW-field has been unequivocally stablished[49,59]. We foresee new exciting avenues of theoretical work along the hydrodynamic crystal concept; the crystalline field theory and its quantized structure among them.

The organizing role of the scale invariances existent within the Faraday harmonics driven upon parametric resonance, and the leading influence of surface stiffness on the nonlinear FW-interaction can be envisaged at the core of a forthcoming scalar field theory based on the Lifshitz's action[50], and its higher-order generalizations for highly correlated systems[60,61]. As a wet-lab realization of the Feynman's simulator[62], our hydrodynamic crystals could result in useful to manipulate 2D-matter waves at interaction with different materials and external fields. Hypothe-tically, quantized coherence and discrete-state superposition can be encoded in hydrodynamic crystals at room temperature, thus offering promise as classical-limit Feynman's emulators for gen-uine quantum effects akin to quantum particle condensation[20,63], and quantum computing with qubits of trapped-ions[64], much more difficult to handle at ultracold temperatures.

To summarize, our elegant hydrodynamic crystals not only constitute an innovation for future developments in material synthesis with soft and biological matter but also a novel arche-type of waving ordering in two dimensions, which could shed light on the interactions between matter and waves at the critical

crossover from the weak hydrodynamic turbulence to the Faraday regimes where wave order emerges as a crystal in two dimensions.

## Methods

**Liquids and suspensions.** Ultrapure water was from a Milli-Q source (Millipore). All the chemicals and surfactants used, including β-aescin, were from Sigma. For improving surface visualization, highly diluted aqueous suspensions of PS-MAA and PS microparticle were synthetized by surfactant-free emulsion polymerization (see Supplementary Note 4).

**Parametric excitation of gravity-capillary waves.** Liquid surfaces under external forcing at a single frequency $\omega_0$ respond as an ensemble of nonlinear surface waves (NLSWs) of master wavelength $\lambda_0$, which defines the fundamental mode determined by a natural frequency equalized to the forcing frequency $\omega_0$. In the linear regime (Hookean), only the fundamental response appears as a monochromatic gravity-capillary wave restored by gravity and surface tension. In the deep-water approximation, the gravity-capillary dispersion relationship is given as $\omega^2 = gk + \sigma k^3/\rho$ where $g$ is the acceleration of gravity and $k = 2\pi/\lambda$ the wavevector; $\sigma$ and $\rho$ accounts, respectively, for the liquid surface tension and density[65]. At frequencies lower than the capillary frequency $\omega_c = (\rho g^3/\sigma)^{1/4}$, one founds GWs of large wavelength controlled by the acceleration of gravity, this is $\lambda_g \approx 2\pi g/\omega^2$ at $\omega < \omega_c$; in water, $\omega_c \approx 14$ Hz and $\lambda_g \geq 1$ cm. At higher frequencies above the capillary frequency ($\omega > \omega_c$), a crossover occurs to the capillary regime dominated by surface tension; here, the CWs have a shorter wavelength ($\lambda_c \ll 7$ mm) varying with frequency as $\lambda_c \approx 2\pi\sigma/(\rho\omega^2)^{1/3}$. The presence of in-plane rigidity due to a viscoelastic film causes the capillary waves to propagate at a slower velocity characterized by smaller wavelengths than in the bare surface[66]. We have deduced a perturbative expansion for the elasticity-dependent gravity-capillary dispersion relation in the relevant limit of dominating rigidity, i.e., at $G/\sigma \gg 1$ (see Supplementary Note 5). Nonlinear GC-waves can be also excited in different NLSW-domains[38], which can be tuned as a parametric excitation in terms of the driving amplitude ($A$)[4]. At low amplitudes below a Hookean threshold ($A < A_0$), only the linear response corresponding to the fundamental GC-mode excited at $\omega = \omega_0$ is found. At high amplitudes ($A > A_0$), NLSWs emerge as cascades of resonant harmonics of the fundamental GC-response ($\omega_n = n\omega_0$ with $n = 2, 3,\ldots$)[67]. Energy and symmetry conservation impose all the nonlinear harmonics with a same wave dispersion than the fundamental driving mode[38].

**Surface wave exciter.** We employed a cylindrical vessel fabricated in Plexiglas® with a design optimized to minimize external vibration and meniscus effects. In order to assure waving patterns with a large aspect-ratio, we designed the vessel with dimensions large enough with respect to the typical capillary wavelengths (diameter $D = 20$ cm; height $h = 2$ cm, thus $D, h \gg \lambda \leq 2$ mm). Vertical vibrations of frequency $\omega_0$ are induced by a louder connected to a power amplifier driven by a function generator (Agilent 33220A). A band-pass filter (Stanford SR560) is used to prevent for spurious harmonics in the driving signal; this guarantees monochromatic excitation as $z(t) = A\cos(\omega_0 t)$, where $A$ is the surface vertical displacement of the container. Gravity-capillary waves are spatially damped along a characteristic length that depends on the ratio of the surface stresses and the bulk friction. The largest decay length is determined by the longest gravity mode, i.e. $l_g \approx \rho g(\lambda/2)A/4\omega\eta \approx g^2A/4\mu\omega^3$; in water (with a kinematic viscosity $\mu \approx 2 \times 10^{-6}$ m$^2$ s$^{-1}$), for the largest vertical amplitudes ($A \leq 2$ mm), elicited at the slowest driving frequencies (ca. 10 Hz), one estimates $l_g \leq 10$ cm, which fixes an adequate vessel diameter at $D = 20$ cm $\gg l_g$. The oscillatory liquid displacements drive the surface against restoring gravity and capillary forces at acceleration uniquely determined by the excitation characteristics as $a = A\omega_0^2$. Only a limited $a$-range was experimentally accessible below the capillary frequency, since for $\omega_0 \ll \omega_c$, one has $a \ll g$. In order to avoid vibration parasites, the experiments were placed on an antivibration table (Newport).

**Laser doppler vibrometry.** Surface vertical displacements were detected by a laser Doppler vibrometer (Polytec® PDV100). A HeNe laser beam (632.8 nm; 1 mW) is focused into a surface spot placed in the center of the liquid vessel. The reflected signal is analyzed in an interferometric detection scheme to retrieve the surface normal velocity in the time domain. The system resolution allows vibrational velocity measurements at $0.02\ \mu$m s$^{-1}$ accuracy with precise linearity across the entire dynamic range in velocity measurement (>90 dB), which allows power measurements as low as $10^{-16}$ m$^2$ Hz$^{-1}$, nominally varying by a factor larger than $10^{12}$. Data acquisition is performed in real time at 22 kHz readout. The analogic signal is DA converted (24 bit) and PC transferred via integrated VibroLink® connector and data cable (Ethernet).

**Spectral analysis.** The LDV setup allows spectral investigations in a broadband spectral range (from 10 Hz up to 22 kHz) for a large span of powers as routinely calibrated against the driving signal; from the practical detection bottom at $10^{-16}$ m$^2$ Hz$^{-1}$ compatible with measurement noise up to millimetric displacements. The power spectral density (PSD) is determined by FFT of the time series of surface velocities acquired by LDV (using VibSoft® 5.5.1). The spectral amplitudes $\psi_n$'s

are determined as the peak maxima measured for the detected harmonics $\omega_n$'s. The spectral exponents are determined as power laws for the power spectral densities PSD$_n \equiv \psi_n^2(\omega_n)/\omega_n \sim \omega_n^{-\alpha}$ (determined as the slopes for the linear fits to the log-log plots). At least five different spectra were recorded at every experimental condition; the experimental slopes correspond to the statistical average $\alpha \pm \sigma_\alpha$ ($N \geq 5$).

**Surface wave imaging.** The vertical acceleration ($a$) was determined as an averaged value of the derivative of the surface velocities measured by LDV at different displacements. Negligible water evaporation, neither changes in surface tension nor temperature drifts occurred during the experiments. Optical imaging was performed with a CCD camera (Nikon D5600) under white light illumination of the water surface. To enhance optical contrast a few droplets of PS-MMA suspension were dissolved till milky shadowing of the liquid surface.

**Interfacial rheology.** We employed a hybrid rheometer (TA instruments, DHR20) working in the interfacial mode with a ring tool oscillating at variable frequency ranging 0.1–100 Hz and a fixed 0.1% shear strain in which the surface response is completely linear (see Supplementary Note 6). We consider the shear rigidity as the in-phase response measured at 1 Hz in the linear regime as an average for at least three curves. Typically, the shear modulus is affected by 20% standard deviation.

## Data availability
Source datasets containing NLSW-spectra are provided as Excel files with DOI 10.6084/m9.figshare.13487541.v1. The files are available at FigShare (free accession at https://figshare.com/articles/dataset/Hydrodynamic_crystals_2D_Faraday_waves/13487541/1).

## Code availability
All source codes are available upon reasonable request to the authors.

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

## Acknowledgements

We thank M.G. Velarde and J. Santamaria for fruitful discussions. The authors acknowledge Comunidad de Madrid for funding this research under grants Y2018/BIO-5207 and S2018/NMT-4389, and Ministerio de Ciencia e Innovación (MICINN) under grant PID2019-108391RB-I00. We thank our colleague Dr. E. Montoya, who provided insight and expertise with the LDV device, and Prof. J. Fernández-Castillo for generosity in free leasing laboratory space.

## Author contributions

F.M. designed the research, provided the experimental methods and discussed the results. M.K. performed the experiments, analyzed the data and discussed the results. N.C. and H.L.M. analyzed the data and discussed the results. E.E. provided materials and discussed the results. J.A.S. and F.M. developed theory. D.H.A. elaborated informatic codes and performed simulations. M.K., N.C. and F.M. wrote the paper and designed and realized the Figures.

## Competing interests

The authors declare no competing interests.
