## [Peer Review File · Nature Communications]

REVIEWER COMMENTS

Reviewer #1 (Remarks to the Author):

By Mikheil Kharbedia, Niccolo Caselli, Horacio Lopez-Menendez, Eduardo Enciso and Francisco Monroy. The authors present their experimental study of parametrically excited Faraday waves (FW) on a liquid surface in a vertically oscillating container of the diameter 20 cm and depth 2 cm. Their interest to a well-known FW phenomenon is focused on a new coherent wave order states related to surface rigidity induced and increased by an added surfactant. As far as I know, the effect of rigidity of the surfactant film on surface waves had not been investigated before and this study is unique.

The manuscript is well structured. It consists of Introduction, Results divided by five subsections, Discussion, Outlook, and Material and Methods parts. The first and second parts of the Results present images and spectra similar to already known from previous studies. There are questions about the spectral slopes shown in Fig. 2(a). What is the interpretation of the spectral parts with the slopes -5 and -2? I would also ask authors to check the statement about spectral slopes $-17/6$ and -5. The range of spectral amplitudes are missing in 2 of 4 plots in Fig. 2(a), and what is shown (from 10-4 to 10-16) looks excessive, which need to be explained.

The further three parts of the Results section describe the pioneering new observations of the hydrodynamic 2D-crystal formation.

The concluding chapters of the manuscript, Discussion and Outlook, adequately present the practical perspectives of hydrodynamics crystals in synthesis of biomaterials from macro- to nanoscales.

I would like to emphasize that the list of references in the manuscript represents quite a wide range of related previous fundamental and applied results.

I recommend this manuscript for publication in Nature Communications after some corrections mentioned above.

Reviewer #2 (Remarks to the Author):

The manuscript "Moulding hydrodynamic 2D-crystals upon parametric Faraday waves in shear-functionalized water surfaces" considers a soft elastic layer on the top surface of a vibrated fluid. So-called Faraday waves commonly appear in a fluid shaken at a resonant frequency, but do not in general exhibit crystalline order. By contrast, the authors report that the shear rigidity of the elastic layer leads to regular crystalline patterns.

I find that the manuscript includes some interesting results. For example, the authors emphasize that the 2D crystals appear independent of container shape and do not rely on friction, unlike other examples of regular 2D patterns in vibrated fluids. However, I find that parts of the manuscript are difficult to read. Before I can make a recommendation regarding the publication of this article, I suggest that the authors rewrite some of the text, including addressing the points detailed below.

- The introduction and discussion are especially difficult to understand, including phrases such as "modernly envisaged," "phase-randomized turbulent cascades of harmonics," "hydrodynamic skeleton," "FW-supported hydrodynamic crystals," etc... I suggest that the authors carefully define any new terminology that they introduce and explain in detail any concept such as the connection with "de Broglie duality" and "Feynman's simulator" that they rely on. The paragraph starting with "As a material condition" could be completely rewritten. In Fig. 2, reference is made to subparts (i), (ii), etc but these are not labeled.

- Scientifically, the authors could better explain the mechanisms behind their results. Why does an

elastic layer lead to more regular 2D patterns? How is the Kolmogorov-Zakharov cascade relevant to these results? I did not follow the connection between the figures measuring these power spectra and the discussion of the elastic layer leading to 2D crystals.

POINT-BY-POINT RESPONSE TO REVIEWER COMMENTS

Moulding hydrodynamic 2D-crystals upon parametric Faraday waves in shear-functionalized water surfaces, by Kharbedia et al.

REVIEWERS' COMMENTS in blue

AUTHORS ANSWERS in black

Reviewer #1

The authors present their experimental study of parametrically excited Faraday waves (FW) on a liquid surface in a vertically oscillating container of the diameter 20 cm and depth 2 cm. Their interest to a well-known FW phenomenon is focused on a new coherent wave order states related to surface rigidity induced and increased by an added surfactant. As far as I know, the effect of rigidity of the surfactant film on surface waves had not been investigated before and this study is unique. The manuscript is well structured. It consists of Introduction, Results divided by five subsections, Discussion, Outlook, and Material and Methods parts. The first and second parts of the Results present images and spectra similar to already known from previous studies.

We gratefully thank the referee for the careful examination and the very constructive that led us to develop a revised version of the manuscript with a stronger physical interpretation of the experimental results. Stimulated by the questions and comments about the spectral cascade, along with Reviewer #2 suggestion to explain the mechanisms behind the observations, we have elaborated a revision that strengthens the physics underneath our problem. Using the general definitions and conventional concepts of spectrum analytics, we have given specific account for the detected spectral features in a plausible connection with the physical mechanisms behind the observed phenomena.

The specific points have been addressed in the manuscript, and especially into the new supplementary materials from which a picture emerges on the Faraday wave (FW) crystallization phenomenon as a coherent case of discretized surface wave tightly coupled to bulk inertia. We very much appreciate your discernment on the uniqueness of this work on hydrodynamic crystal formation and the new physics it could represent, so we hope to have enhanced the clearness and meaningfulness of the revised version in accordance.

There are questions about the spectral slopes shown in Fig. 2(a). What is the interpretation of the spectral parts with the slopes -5 and -2? I would also ask authors to check the statement about spectral slopes -17/6 and -5.

We especially acknowledge these specific questions as being at the core of the concerned physical problem. The spectral slope -5, understood as a dynamical signature for FWs excited upon liquid vibration, has been previously reported in at

least two experimental papers (Punzmann et al. PRL 2009; Snouck et al. Phys. Fluids 2009). To the best of our knowledge, however, no mechanistic account for the (-5) FW-distinctive has been delivered yet. In our opinion, the -5 slope appears as a consequence of the FW-resonances leading the wave interactions that give rise to a more ordered wavefield than in the KZ- (pre-FW) regime. Indeed, the -5 signature also disappears upon further disordering excitation in the chaotic (post-FW) state. Therefore, it should be explained by reference to the $-17/6$ (or -4) slopes observed in the KZ-states, and also in regard to the -2 slope, representative for unsteady Landau's chaos realized in our experiments as a continuous superposition of modes at increasing excitation. As being crucial for a physical understanding at the core of the FW-freezing phenomenon (but conscious of the huge complexity of the problem), we have made the exercise to explain these phenomenological changes of spectral characteristics in the most unified theoretical framework as possible. Two revision scenarios, minimal and extended, are proposed.

Minimal revision. In a mechanistic regard to harmonic oscillators in one dimension, the -5 slope is expected to appear as the tail for the power spectral densities of a monochromatically forced oscillator at balanced resonance with an inert mass. This spectral feature was argued for turbulence of surface waves at equilibrium with forcing wind (Phillips-like spectrum PSD $\sim \omega^{-5}$), in which a detailed balance between driving force and waving mass is assumed for resonance at each frequency. Further consideration of viscous friction should give rise to a terminal tail (PSD $\sim \omega^{-3}$). Although the terminal slope -3 was not explicitly mentioned in the first manuscript, it is systematically observed at high frequencies and amplitudes (at high rates indeed, as corresponds to a frictional death). Therefore, inertial resonance with a -5 spectral envelope (and frictional -3 spectral tail) can be reasonably assumed as the driving force that lead FW formation (and frictional death) at resonant balance with the monochromatic source of driving. Further, intrinsic surface rigidity should be assigned as the specific interaction at the core of the FW-crystallization phenomenon.

The revised Fig. 2 and related texts have been corrected in accordance to this mechanistic picture quoted in the revised version as distinctive features for surface FW-resonance under inertial (-5), or frictional (-3) governance. To better explicit the connection between the experimental spectra and such a plausible theory, we have revised the introduction, included new references, and better explained the introductory section of experimental results by invoking the underlying concepts of resonance and resonant interactions. The piece of discussion above depicted would be the minimal essential to be considered in a minimally modified revised version. An extensive argumentation including exact analytics and numerical simulations has been elaborated in the new supplementary materials (see below).

Comprehensive revision. In addition to the explanation for the particular $-5/-3$ spectral slopes predicted from the theory of forced oscillators (as exact analytic solutions), a unified physical framework has been constructed in Supplementary Note N1 to give comprehensive account for the different regimes of nonlinear surface waving (NLSW). We have pivoted on the forced nonlinear Duffing oscillator

(DNLO), which includes inertia (FW-forming) and an intrinsic stiffening-parameter that accounts for tuning the wavefield self-focusing involved in FW freezing; the one dimensional, DNLO- equivalent describes (at increasing forcing):

a) The KZ-regimes of disorganized surface waving creating nonlinear harmonics under weak interaction at lower amplitudes (-4 for GWs, and $-17/6$ for CWs). They are further interpreted in Suppl. Note N2 as cases of dominant wave dispersion leading to intrinsic resonances with a spatial structure describable by the diffusive structure of the nonlinear Schrödinger equation (a special case of DNLO with developed spatial structure but devoid of inertia; as seminally identified by Zakharov and cols. Physica D 2001; Phys. Rep 2004).

b) The FW regime of parametric resonance when bulk inertia starts to be influential. From the DNLO-framework, we describe numerically composite cascades of harmonics and subharmonics with the distinctive slope-5 observed in experiments. The FW-DNLO cascades are predicted to end in a terminal domain of slope -3, as expected for a frictional death. Theory shows FWs emerging naturally as the steady solutions of subharmonic DNLO-resonances appeared under monochromatic AC-forcing at non-negligible inertia. Duffing stiffening is shown to lead the wavefield self-focusing needed to lock the FW phases. From these numerical results, a plausible coherence mechanism naturally arises for FW-freezing as steady patterns with a permanent form (hydrodynamic crystals).

c) The chaotic regime of continuous wave turbulence emerged as molten FWs at higher amplitudes. The DNLO predicts a FW-chaos in which more and more subharmonic modes fill the frequencies in a near-continuum spectrum with a Lorentzian envelope. We explain this feature as a Landau's superposition of unsteady modes progressively realized under subharmonic resonance of the formers. The w^{-2} feature is dictated by the exponential dynamics of growing at turbulent unsteadiness (at Lorentzian spectral profile), which is limited by frictional death in the faster modes (decaying as w^{-3}), as observed in the experiments.

The range of spectral amplitudes are missing in 2 of 4 plots in Fig. 2(a), and what is shown (from 10^{-4} to 10^{-16}) looks excessive, which need to be explained.

Figure 2a has been corrected by adding the missing labels in the PSD axes and made explicit their units at m^2 / Hz . Additional details on spectrum LDV-measurement and spectral analysis have been included in the Methods section. The available logarithmic span of powers stands on the broad dynamic range of the LDV vibrometer at linear accuracy of velocity measurement (> 90 dB). Measure is performed in time domain at 22 kHz bandwidth readout. The analogic signal is D/A converted (24 bits), and PC transferred at maximal broadband via integrated VibroLink® connector. These instrument settings allow for logarithmic expansion by more than ten decades in power and by three decades in frequency domain. This dynamic range is routinely calibrated by direct LDV measurements in a mirror surface attached to the driving louder. Therefore, a 10-10 span ratio in power measurement is practicable starting from a noise bottom as low as $10^{-15} m^2 / Hz$ up to $10^{-5} m^2 / Hz$ (or higher at saturation). Similar measurements with LDV vibrometers working at comparable power ratio in a lower frequency band can be

also found in literature; see e.g. Blamey et al. Langmuir 29, 3835 (2013); Issenmann et al. EPL 116, 64005 (2016); Michel et al. Phys. Rev. Fluids 3, 054801 (2018).

The further three parts of the Results section describe the pioneering new observations of the hydrodynamic 2D-crystal formation.

The concluding chapters of the manuscript, Discussion and Outlook, adequately present the practical perspectives of hydrodynamics crystals in synthesis of biomaterials from macro- to nanoscales.

I would like to emphasize that the list of references in the manuscript represents quite a wide range of related previous fundamental and applied results.

I recommend this manuscript for publication in Nature Communications after some corrections mentioned above.

We thank again the referee for the favorable opinion on our work, and hope you find the revised version and supplementary materials to mean a significant improvement with respect to the former version of the manuscript.

Reviewer #2 (Remarks to the Author):

The manuscript "Moulding hydrodynamic 2D-crystals upon parametric Faraday waves in shear-functionalized water surfaces" considers a soft elastic layer on the top surface of a vibrated fluid. So-called Faraday waves commonly appear in a fluid shaken at a resonant frequency, but do not in general exhibit crystalline order. By contrast, the authors report that the shear rigidity of the elastic layer leads to regular crystalline patterns.

I find that the manuscript includes some interesting results. For example, the authors emphasize that the 2D crystals appear independent of container shape and do not rely on friction, unlike other examples of regular 2D patterns in vibrated fluids. However, I find that parts of the manuscript are difficult to read. Before I can make a recommendation regarding the publication of this article, I suggest that the authors rewrite some of the text, including addressing the points detailed below.

We gratefully thank the referee for the clever judgment on the need for better explaining the physics behind the observed wave patterns and the role of shear rigidity. A substantial revision has been carried out in consequence, pursuing two main aspects: 1) improving text readability by rewriting the obscure points indicated; 2) including new supplementary materials with the mechanistic interpretation behind the observable phenomenon. These changes detailed below.

- The introduction and discussion are especially difficult to understand, including phrases such as "modernly envisaged," "phase-randomized turbulent cascades of harmonics," "hydrodynamic skeleton," "FW-supported hydrodynamic crystals," etc... I suggest that the authors carefully define any new terminology that they

introduce and explain in detail any concept such as the connection with "de Broglie duality" and "Feynman's simulator" that they rely on. The paragraph starting with "As a material condition" could be completely rewritten. In Fig. 2, reference is made to subparts (i), (ii), etc but these are not labeled.

Thank you very much for the careful examination of the manuscript; the comments have been certainly useful for improving the revised version in the specific points indicated, and in general too. New supplementary materials have been elaborated to expand on the concepts invoked, to introduce unconventional terminology, and fundamentally, to enlighten the physics behind the observed phenomena in a theoretical perspective. They describe surface motion as a dynamic system represented by a nonlinear Duffing oscillator (the DNLO); exact analytic results and numerical simulations based on nonlinear oscillators are presented. Although we refer you to the specific contents for examination, you will find below a claim on how they contribute to understand the physics concerned. Following the suggestions about improvements in the main text, the specific changes are:

- **Introduction.** The second paragraph has been significantly revised. We have reintroduced the concept of "hydrodynamic skeleton" by reference to the deterministic dynamics that governs the internal organization of fluid flows. A quote to the most relevant papers on fluid mechanics of two-dimensional flows has been included by reference to this concept. In addition to the previous references on two-dimensional flows (Refs. 21, 23, 26, 29-32), we have included the foundational paper of Zakharov on the Hamiltonian formulation of water wave turbulence (new Ref. 22), the review of Tabelling in Physics Reports on two-dimensional turbulence (new Ref. 24), and the paper by Falkovich in Physics Today (new Ref. 25), in which the current concept of "hydrodynamic skeleton" was explicitly introduced perhaps for the first time. The concept of "resonant interactions", and the role of "harmonic nonlinear resonance" as key factors of surface wave organization have been made explicit by reference to the classical review by Hammack and Henderson (new Ref. 27). Regarding FW formation, the concept of parametric resonance appears now better explained and referenced, having included the review by Milles and Henderson (new Ref. 3). The specific role of extrinsic resonance in Faraday wave formation appears now explicated by reference to the paper by Muller et al. in PRL 1997 (new Ref. 28).

- **Discussion.** A new paragraph has been added to integrate the theoretical perspective contributed from the new supplementary materials. They have been organized along three supplementary notes, which specifically deal with:

Suppl. Note N1. *Resonance status of nonlinear water surface wavefields (NLSW): hydrodynamic regimes and spectral signatures.* A minimal description of the surface wavefield is introduced in terms of nonlinear oscillators, particularly the Duffing oscillator (DNLO), which captures the idea of surface rigidity as a nonlinear stiffening that self-focuses the field to achieve the wave coherence necessary for

the FWs to freeze as “hydrodynamic crystals”. The concept of extrinsic resonance of the bulk liquid with the driving source is introduced as the dynamic skeleton able to scaffold the systemic response in a discretized structure of nonlinear harmonics. The idea of harmonic response as a wavefield generator, and the subharmonic response as a piloting field for the FW, are also introduced. The physics of the energy cascades is depicted from the DNLO not only for the main harmonic field but also for the parametric resonator (FWs), which is recovered in the Mathieu equation for the subharmonic FW-like response. Because no space is available in the main text for further details on this physics, we have exploited Suppl. Note N1 to introduce the main ingredients and the concepts needed to understand the observed phenomenology from the DNLO, to explain the new physics that relies on, and as a main goal, to describe in detail the specific spectral features of each one of the hydrodynamic regimes observed in the experiments. From this simple model, the Suppl. Note N1 focuses on the physical meaning of the observed spectral slopes (as also requested by Reviewer #1).

Suppl. Note 2. *Nonlinear water surface wavefields: DNLO unification.* Building upon the results in Suppl. Note N1 about nonlinear oscillators, a unified wisdom of the different classes of surface wavefields describable by equations of motion of the DNLO-class is given in this Suppl. Note N2. As a main goal, a conceptual map is plotted as a schematic for a common equation of motion (MMT framework), which recapitulates the different classes of surface wavefield. By neglecting one or several physical components of the MMT, we recover the different domains of surface waving motion observed upon increasing forcing; from KZ-disorganized motion, through of FWs able to undergo self-focusing under enough stiffening, up to the Landau’s regime of unsteady turbulence giving rise to chaotic waves. From this unification, a central role emerges for bulk inertia as the discretizing ingredient that enables the partition of space and time into discrete pieces by extrinsic resonance with the external source of driving (monochromatic). Then, bulk inertia together with surface rigidity, appear as the chief components of the “hydrodynamic skeleton” that support the infrastructure of the hydrodynamic crystals.

Suppl. Note 3. *Towards a classical field theory of hydrodynamic crystals: Faraday matter waves.* Finally, in a separated piece of theoretical physics, we depict the elements that could pave the way towards a classical theory of the discovered hydrodynamic crystals. The concept of inertia-force discretization under resonance is introduced as a starting point to reorganize the equation of motion as a discretized wave equation of the Klein-Gordon (KG) class. The inertia appears then together with rigidity as the two systemic generators at the core of a scalar potential that describes the forces in a discretized waving system, which can only operate by integer units of a “quantum of action” as defined in terms of the discrete pieces of mass, length and time imposed by the discrete nature of the resonance with the monochromatic source. Consequently, a discretized momentum-energy

relationship appears, from which an equivalent de Broglie relationship is obtained between the momentum of the FW and its wavelength, in terms of the “quantum of action”, which appears classically discretized due to the discrete nature of the systemic excitation.

Therefore, the new concepts and the apparently foreign analogies invoked in the discussion and outlook sections, including the “De Broglie duality” and the “Feynman simulator” would now appear more naturally as quantum alike concepts in the classical sense of discretization. After calling to them from the new materials included as Supplementary Notes N1-N3, the claims to these concepts have been better nuanced in the main text as genuine classical realizations of a discretized system. Of course, the paragraph starting with "As a material condition" has been rewritten to enhance readability. In Fig. 2, reference to subparts i), ii), iii) and iv) has been made explicit.

- Scientifically, the authors could better explain the mechanisms behind their results. Why does an elastic layer lead to more regular 2D patterns? How is the Kolmogorov-Zakharov cascade relevant to these results? I did not follow the connection between the figures measuring these power spectra and the discussion of the elastic layer leading to 2D crystals.

You are absolutely right; indeed, we strongly acknowledge this crucial concern (which is ours too) on the need to reveal the mechanisms behind the observed phenomenon. As a general answer to your point of concern, there is a consensus among FW theorists on the parametric oscillator character of the fluid surface at subharmonic resonance of the bulk fluid with the driving force (Miles & Henderson 1990; Rajchenbach and Clamond 2015). In Suppl. Note N1, we specifically describe how liquid inertia and surface wavefield self-interaction cooperate in the DNLO to drive the parametric resonance that gives rise to FWs in one dimension. The characteristically amplified parametric subharmonic response encoded in the DNLO is inherent to the observed FWs in vibrated liquids being thus an essential behavioral trait at the physical core behind the FW-crystallization phenomenon. Such an extrinsic resonance gives account for the -5 spectral feature and the phase locking observed under field stiffening, two facts extensively described and justified in Suppl. Note N1. Further, the state of the art in FW physics has been comprehensively incorporated to the relevant elements considered in this supplementary piece of theoretical work.

The invoked analogies between the hydrodynamic nonlinear surface wavefields (NLSW) and the dynamic DNLO have been reasonably justified, particularly regarding the FWs. From the mathematically rigorous description of FWs as the natural response of liquids in vertical periodic motion (Benjamin & Ursell 1954), the central role of bulk inertia is well established in the FW analytics (see e.g. the Miles & Henderson review, or Muller et al. PRL 1997). Most theoretical investigations describe FW subharmonic physics as drawn by Mathieu oscillators with the natural

elastic response of the surface modulated by restoring forces in a frame of reference comoving with the container (a quote to this description is made in the new Ref. 28). By using the DNLO as a minimal description of the ordinary harmonic wavefield superposed to the extraordinary subharmonics, we have therefor recapitulated bulk inertia and elastic surface response in a minimal nonlinear paradigm that reasonably recapitulates the physics behind our results. Although the first version was merely descriptive of the experimental evidence in a quite phenomenological basis, your sharp comment has encouraged us to enroll a much deeper physical understanding in a theoretical basis. In consequence, we have augmented the paper with the three supplementary pieces of theoretical work above enumerated. Regarding your specific questions from the phenomenology after the theoretical analysis performed, we know hitherto:

Why does an elastic layer lead to more regular 2D patterns?

We have shown that lateral rigidity causes wavefield self-interaction leading higher and more coherent parametric resonance. Thus, stiffening driven wavefield self-focusing should be the interaction that pilots the locked phase coherence observed in the experiments as the structural cause for wave freezing in a steady pattern with a permanent form. The physics of the resonant mass-spring system rigidified under self-focusing stiffening is encoded in the system representing DNLO (as explained in Suppl. Note N1). Further, it can be made compatible with the spatial structure inherited from the previous KZ-regime (as encoded by the diffusive structure of the nonlinear Schrödinger (NLS) equation; see Suppl. Note N2). Finally, layer elasticity can be also implemented as a quartic scalar self-interaction together with the mass-spring at the kernel of the proposed theory of hydrodynamic crystals (Suppl. Note N3). These different theoretical perspectives emphasize on the ordering role of the surface layer rigidity, which appears as the chief factor for the surface wave “crystallization” observed in the experiments.

How is the Kolmogorov-Zakharov cascade relevant to these results?

It represents the pre-FW state, which imposes the intrinsic hydrodynamic skeleton of internal resonances compatible with wave dispersion (four-wave coupling for GWs and three-waves interaction for CWs). We have also shown how this structure is encoded in the NLS equation and why it is primarily inherited by the FW generating field (the cascade of ordinary harmonics). The FW-field (the extraordinary cascade of subharmonics) pilots the parametric resonance, which is amplified, intrinsically self-focused, and practically inviscid, thus very supportive for the inertial crystal structure (as observed in the experiments). As encoded in the DNLO through the nonlinear term, a significant coupling between both wavefields could be likely for maintaining the frozen structure. All this physics is discussed in Suppl. Note N1. In addition, the ascendant influence of the KZ-field can be now derived as natural from the unified wisdom included in Suppl. Note N2. Furthermore, the diffusive structure imposed by the NLS equation (at the core of

the KZ-description) is also inherited by the Lifshitz field proposed to describe the hydrodynamic crystals in Suppl. Note N3. All these relevant connections are now better understandable from the explicit materials proposed.

I did not follow the connection between the figures measuring these power spectra and the discussion of the elastic layer leading to 2D crystals.

Gaining phase locking (coherence) upon enough wavefield stiffening is the critical point for FW freezing (together with the FW-distinctive spectral slope -5, which has been theoretically identified as the signature for balanced extrinsic resonance between external source and liquid inertia). This physics can be easily understood by invoking the DNLO equivalent analyzed in Suppl. Note N1. The specific wavefield features leading FW freezing can be clearly identified in the experimental spectra, being now better understood in terms of coherent resonances as encoded in the DNLO equivalent. Because the results' section has been enhanced with the theoretical perspective provided by the DNLO equivalent (by reference to the theoretical results in Suppl. Note N1), we hope the revised discussion of the elastic layer at clearer comprehensibility than in the first version. Furthermore, an explicit mention to surface stiffening as the main promoter of spatial correlations and dynamic steadiness has been included in the description of the crystallization data in Fig. 5 (by reference to the new supplementary materials).

Editor

As you will see from the reports copied below, the reviewers raise important concerns. We find that these concerns limit the strength of the study, and therefore we ask you to address them with additional work. Without substantial revisions, we will be unlikely to send the paper back to review.

We are very grateful for the opportunity you have offered us to send the paper back to review in case a substantial revision with additional work is carried out. We have understood such a substantial revision as a comprehensive account of the physics behind the results together with an explication of the physical mechanisms for hydrodynamic crystallization. Otherwise understood, the phenomenon itself is not enough so that a physical insight is also required. Very much encouraged by the reviewer's comments and by your invitation to revise, we have worked out on such a physical conceptualization. To perform such an effort a theorist has been enrolled (Prof. JA Santiago) and an expert in numerical simulation (Prof. D Herráez-Aguilar). The new collaborators have significantly contributed to build the theoretical edifice that would mean a strength for overcoming the substantial concerns raised by the reviewers on the need to explain the physics behind the problem. Although you have already found a detailed explanation for the physics covered by the new supplementary materials in the answers to the reviewers, here

we more concisely claim how the additional work could contribute to strength our study.

Suppl. Note N1. Looking at the observed phenomenology, we have constructed in a comprehensive model of nonlinear oscillators that captures the several phenomena appeared and the physical ingredients involved in the course of our experiments. A plausible description of the hydrodynamic regimes and the spectral features arises. This physics, being quite conventional, is rather descriptive and quite predictive of the observed phenomenology, at least in qualitative terms. We consider this modulus to contain the essential insight to be considered for a final version in case.

Suppl. Note N2. The relevant interactions have been synthesized into an analytic program with a global predictive value not only for the dynamic features captured from the spectra (amplitudes, phases, cut-off frequencies, etc.), but also for the trajectories and phase portraits that can be also compared with experiments performed in different states (including the hydrodynamic crystals). This physics is plausibly unifying and quantitatively predictive in a broad parametric space that recapitulates most of the observable features under a unified theory. We consider this modulus very important to realize a global understanding of the phenomena.

Suppl. Note N3. This piece of theoretical work depicts the hypothetical preliminaries of the forthcoming field theory of the hydrodynamic crystals. However, this preliminary physics would be enough informative, and useful to construct de novo a scalar field theory from a complete axiomatics based on the observed phenomenology. We consider this piece in principle accessorial, although pretty much informative for theorists interested to boost the new physics in this direction. If discarded, the outlooking claims of hydrodynamic crystals for potential applications as matter waves or in future developments as a classical Feynman simulator could seem unsupported.

As far all your points of concern are aligned on the need to provide better physical insight behind the observed phenomena, we consider all these theoretical analytics as an adequate, comprehensive proposal for the substantial revision demanded. Not only the detail for the spectral slopes but also the concept for “FW crystallization” upon coherent phase-locking, and its melting into incoherently waving relatives emerge naturally all of them from the DNLO model. The new supplementary materials give a plausible explanation for the observed phenomena, particularly on how the FW patterns can be entailed upon surface stiffening among the different classes of surface waving (observed each one with a distinctive spectral profile at increasing forcing; Suppl. Note N1). The quantitative study with DNLO (theory and numerics), has been integrated together with the unifying physics that covers the transitions between the corresponding waving states (Suppl. Note N2), and a tentative depiction for a forthcoming field theory of the new “hydrodynamic crystal” state (Suppl. Note N3). The phenomenological evidences in

the results' sections, and its discussion as well, have been complemented with the theoretical perspective arisen from the new materials.

In our opinion, the three supplementary pieces of theoretical work provide the adequate complement on the theoretical physics behind the experimental results (although they could be reduced even partially suppressed if necessary). Despite the substantial strengthening provided by the three supplementary pieces of theoretical work, we understand that the package could result too much heavy, so that each one of the pieces can be considered separately to contribute the final version of the work in case it results finally accepted. We have designed these three pieces as being modular and interconnectable, thus endowed with the possibility to be considered separately. Although the three moduli considered together could constitute the substantial work necessary to globally address your concerns, they must be considered as a maximum proposal that integrates the answers to the specific points of the reviewers in a complete edifice giving theoretical support to the observed phenomenon. This effort has made us realize the weakness in the physical narrative of the former version, which we believe now completely strengthened in the revised version including the new supplementary materials. We are confident of having addressed your demand on a substantial revision through the new theoretical work, which points out the actual importance of the discovered phenomenon hopefully at the best satisfaction of the journal's standard.

REVIEWERS' COMMENTS

Reviewer #1 (Remarks to the Author):

I am satisfied with authors answers and revision of the manuscript and recommend the amended version for publication in Nature Communications.

Reviewer #2 (Remarks to the Author):

The authors have addressed my remarks about the scientific content of the article well.

As a minor comment, I find that the clarity of the language in the manuscript could still be much improved to make it more accessible towards the broad readership of Nature Communications. There are many abbreviations, "FW" "NLSW" "NLO" "GC" etc, that make most sentences difficult to understand. Although some jargon that the authors use has been clarified, other jargon has been introduced. What are "dynamic scaffolds," "hydrodynamic skeletons," "dynamical infrastructures," "extrinsic regulation," "quantized" duality, "Lifshitz-class generator"? I suggest the authors define any terms that are not broadly familiar to any physicist.

I find this comment minor and provided that the authors address it, I believe the manuscript is well on the way towards publication.

POINT-BY-POINT RESPONSE TO REVIEWER COMMENTS

Moulding hydrodynamic 2D-crystals upon parametric Faraday waves in shear-functionalized water surfaces, by Kharbedia et al.

REVIEWERS' COMMENTS in blue

AUTHORS ANSWERS in black

REVIEWERS' COMMENTS

Reviewer #1 (Remarks to the Author):

I am satisfied with authors answers and revision of the manuscript and recommend the amended version for publication in Nature Communications.

Answer: We are very much grateful for the very meaningful review, the time spent, and the great care taken by the reviewer in helping us to improve the paper.

Reviewer #2 (Remarks to the Author):

The authors have addressed my remarks about the scientific content of the article well.

As a minor comment, I find that the clarity of the language in the manuscript could still be much improved to make it more accessible towards the broad readership of Nature Communications. There are many abbreviations, "FW" "NLSW" "NLO" "GC" etc, that make most sentences difficult to understand. Although some jargon that the authors use has been clarified, other jargon has been introduced.

Answer: Thank you very much for advising the excessive use of abbreviations along the manuscript. We have thoughtfully revised the text avoiding the wordy use of acronyms and abbreviations in the previous versions. In the search for a more literal wording at easier reading, the explicit substantives have been used instead; particularly, the changes made are:

We have substituted "FW" by "Faraday wave", and "NLSW" by "nonlinear surface wave", when being at the subject of the statements.

The acronyms' phrasing has been limited to the composed substantives used as the object of the statements; for instance, we used "FW-coherence", "NLSW-state", or similar abbreviated constructions only at request of a compact referral to those specific concepts.

The repeated use of "NLO" for "nonlinear oscillator" has been also avoided.

Finally, the abbreviation "GC" has been removed in favour of the explicit adjective "gravity-capillary" for the surface waves of this class. Only the more conventional

acronyms “GWs” or “CWs”, respectively for “gravity waves” and “capillary waves”, have been preserved when inevitable for a repeated use in a same statement or paragraph.

What are "dynamic scaffolds," "hydrodynamic skeletons," "dynamical infrastructures," "extrinsic regulation,"

Answer: Thanks again to tag these unconventional, or “weird” expressions of jargon. In the revised version we have limited the unwise use of these unconventional terminologies. The following changes have been introduced:

- “organizational rules” instead of “dynamic scaffolds”.
- “hydrodynamic interactions” instead of “hydrodynamic skeletons”.
- “dynamical structures”, or “nonlinear interactions” instead of “dynamical infrastructures”.
- “external control” instead of “extrinsic regulation”.
- “duality” instead “quantized duality”, which was manifestly erroneous to refer to the classically discretized duality of Faraday waves as monochromatic waves made of fluid matter.

What is “Lifshitz-class generator”? I suggest the authors define any terms that are not broadly familiar to any physicist.

Answer: Many thanks to remark this inaccuracy of the main text, which appears clarified well by reading the supplementary materials. In the revised version of the manuscript we expect having this mistake corrected. We refer to the “Lifshitz-class generator”, now better said a “Lifshitz-type action generator”, actually as the Lagrangian functional that generates mechanical action for a massive object in a field with a generalized diffusive structure.

In the revised version of the Discussion we now address this point by reference to the original Lifshitz’s paper on the theory of phase transitions leading ordered phases in highly correlated systems (new reference 50). The revised Outlook includes too an additional comment on the specific role of scale invariances (spatial and temporal), and nonlinear interactions (stiffening) as the fundamental physics to be considered in a forthcoming classical theory of hydrodynamic crystals based on the generalized Lifshitz’s framework. Two additional references have been also quoted to this respect (new references 60 and 61).

I find this comment minor and provided that the authors address it, I believe the manuscript is well on the way towards publication.

Answer: We are very much grateful for the very meaningful review, the time spent, and the great care taken by the reviewer in helping us to improve the paper.